# Beyond Obstruction: Evaluating Self-Expandable Metallic Stents (SEMSs) vs. Emergency Surgery for Challenging pT4 Obstructive Colon Cancer: Multicentre Retrospective Study

**DOI:** 10.3390/cancers16234096

**Published:** 2024-12-06

**Authors:** Marta Paniagua García-Señoráns, Carlos Cerdán-Santacruz, Oscar Cano-Valderrama, Inés Aldrey-Cao, Beatriz Andrés-Asenjo, Fernando Pereira-Pérez, Blas Flor-Lorente, Sebastiano Biondo

**Affiliations:** 1Colorectal Surgery Department, Complejo Hospitalario Universitario de Pontevedra, 36071 Pontevedra, Spain; marta.paniagua.garcia.senorans@sergas.es; 2Fundación de Investigación Sanitaria Galicia Sur, 36213 Vigo, Spain; 3Colorectal Surgery Department, Hospital Universitario de la Princesa, 28006 Madrid, Spain; carloscerdansantacruz@hotmail.com; 4Colorectal Surgery Department, Complejo Hospitalario Universitario de Vigo, 36312 Vigo, Spain; 5Colorectal Surgery Department, Complejo Hospitalario Universitario de Ourense, 32005 Ourense, Spain; ines.aldrey.cao@sergas.es; 6Hospital Clínico Universitario de Valladolid, 47011 Valladolid, Spain; beatrizdeandres007@yahoo.es; 7Chief General Surgery Department, Hospital de Fuenlabrada, 28942 Madrid, Spain; fernando.pereira@salud.madrid.org; 8Colorectal Surgery Department, Hospital Universitario y Politécnico la Fe, 46026 Valencia, Spain; blasflor@hotmail.com; 9Bellvitge University Hospital, Department of General and Digestive Surgery, University of Barcelona and IDIBELL, 08907 Barcelona, Spain; sbn.biondo@gmail.com

**Keywords:** colorectal cancer, colon cancer, self-expandable metallic stent, emergency surgery

## Abstract

pT4 colon cancer can present as an obstructive cancer. In these cases, there are two alternatives for treatment: emergency surgery or the use of self-expandable metallic stents, which resolves the obstruction to allow an elective surgery. Comparisons of these two strategies are scarce. Therefore, we analysed a cohort of patients with pT4 colon cancer who presented with obstruction and compared the results of both treatments. With this information, clinicians treating these kinds of patients will be able to give them better treatment.

## 1. Introduction

Up to 30% of colon cancer cases present as an obstruction, even after the establishment of population screening programmes [1,2,3]. Emergency surgery (ES) has long been considered as the gold standard, despite its association with high morbidity and mortality rates [4,5,6,7]. Self-expandable metallic stents (SEMSs) were described as an alternative to emergency surgery in certain situations, mostly for left-sided colon cancer, currently being one of the most popular alternatives to ES for malignant colonic obstruction. However, SEMS placement is still controversial and a matter of research due to poor outcomes reported by early randomized trials, which had to be prematurely cancelled due to unacceptable morbidity [8,9] and worse oncological outcomes for patients treated with SEMSs in relation to high perforation rates [10,11,12,13].

As a result of the progressive mastery of the technique [14] and the standardization of indications, a substantial improvement in the complication rate and immediate clinical results was observed. SEMSs have progressively become established in the acute setting and have been incorporated into guidelines as an initial approach for left-sided obstructive colon cancer management.

European and American guidelines have described indications for stenting as follows: malignant colonic obstruction without signs of perforation (strong recommendation, low-quality evidence), bridge to surgery for a curative malignant left-colonic obstruction within a shared decision-making process (strong recommendation, high-quality evidence), palliation of malignant colonic obstruction (strong recommendation, high-quality evidence) [15,16].

However, despite the favourable short-term results, the prognostic implications for oncological outcomes due to possible microperforation/tumour perforation during SEMS insertion have been a cause of concern since the beginning.

It is well known that pT4 colon tumours, especially those with obstruction, are considered challenging cases from a technical point of view (ES vs. SEMS) in terms of oncological prognosis, since pT4 and obstruction are two of the strongest risk factors for peritoneal and systemic recurrence [17]. Tumour perforation is an additional risk factor in this setting and is one of the reasons that precluded many authors from SEMS use. Taking this into account, the initial management of this specific, high-risk group of patients with pT4 colon cancer when presenting as an obstructive tumour could influence postoperative and oncological results.

The aim of our study is to compare SEMS placement with ES in selected patients with adverse tumour conditions (pT4 tumours and obstructive setting) and to analyse postoperative and oncological outcomes in both groups.

## 2. Materials and Methods

Local Clinical Research Ethics Committee (CREC) approval was obtained (04/21-4398).

This study is a secondary analysis of an original study registered at ClinicalTrials.gov, number NCT05300789, in which oncological outcomes for a subgroup of pT4 colon cancer patients were analysed [18].

This study adheres to the STROBE (Strengthening the Reporting of Observational Studies in Epidemiology) statement.

### 2.1. Design, Patients, and Variables

An observational retrospective multicentre trial was designed. A total of 50 different hospitals enrolled in the project. This study was sponsored by the Spanish Surgical Society (Asociación Española de Cirujanos), both the Colorectal and Peritoneal Surgery subsections.

All consecutive patients operated on for colon cancer (15 cm above the anal verge) with curative intent (elective or emergency surgery) and with pT4 tumours confirmed by pathological reports were included in the initial database. Patients who underwent operations between 2015 and 2017 were considered for this study in order to achieve a minimum follow-up of 3 years. Initial data analysis was performed in 2021. Subsequently, left-sided obstructive colon cancers (distal to the splenic flexure) were selected for the present analysis, considering that proximal locations are not highly recommended for SEMS placement. Various hospitals and subspecialised surgical teams were admitted to participate.

Diagnosis of complete colonic obstruction was based on anamnesis, physical examination, and radiological findings in the abdominal CT scan; special attention was paid to proximal colonic dilation and ileocecal valve sufficiency. Surgeries were divided into left colectomy (left colon), sigmoidectomy (sigmoid colon), Hartman’s procedure (end colostomy), and others.

Exclusion criteria were palliative surgery or incomplete tumour resection (R2), synchronous metastases (systemic, peritoneal), different histological type than adenocarcinoma, loss to follow-up, and relevant data missing information.

Provision of exact details on the equipment, study protocols, CT scan reports, and obstruction criteria was entirely at the discretion of the participating institution. Definitive management (ES versus SEMS) was decided by on-call surgeons at each hospital, based on clinical patients’ conditions and the availability of SEMSs at each institution.

Data were collected by two senior staff members from each participant centre. Different variables were recorded: demographics, preoperative disease data, surgical or stent management characteristics, pathology reports (based on the 8th edition of TNM classification) [19], postoperative outcomes, and oncological follow-up. The variable ‘’free tumour’’ describes tumours without firm adhesions to adjacent viscera, abdominal wall, or peritoneum/retroperitoneum. Postoperative complications were graded in severity levels according to the Clavien–Dindo classification (I-IV) [20]. Oncological treatment schemes and follow-ups were individualised for each patient, following standardised protocols in accordance with current international guidelines [21], based on the best clinical practice. For local or distant recurrence diagnosis, a CT scan was used. Peritoneal metastases were defined either by pathological findings or imaging (CT scan, PET-CT).

### 2.2. Outcome Measures

Postoperative and oncological outcomes were analysed.

Primary outcomes of this study are focused on oncological metrics: recurrence rates (local, peritoneal, and systemic) and survival rates. Disease-free survival (DFS) was defined as the time interval from surgical intervention to documented recurrence or mortality; overall survival (OS) was defined as the time lapse from surgery to death from any cause. Cancer-related mortality was analysed separately.

Secondary outcomes were immediate clinical results in terms of post-colectomy complications (SEMS vs. ES); optimal surgical results were defined as surgery without major complications, postoperative mortality, or stoma creation in each cohort.

### 2.3. Statistical Analysis Methods

Quantitative variables were expressed as mean values with standard deviation and categorical variables as number of patients (percentage). The normality of quantitative variables was tested using the appropriate Shapiro–Wilk test. Univariate analysis was performed utilising Fisher, χ^2^, and Student *t* tests, as appropriate, to assess the association between different independent variables. Statistical significance was determined for differences with *p* < 0.05.

Variables that were statistically or clinically correlated with the treatment group were considered confounding factors and included in the propensity score analysis (PS) (*p* < 0.05, odds ratio (OR) > 1.5, OR < 0.67, Pearson correlation > 0.1, or Pearson correlation < 0.1). Only variables existing at the time of treatment group selection were included to mitigate potential selection bias. All other variables were considered result variables, not adjustable with the PS. A logistic regression model was constructed with the treatment group as a dependent variable and confounding variables as independent variables. Once the logistic model was created, it was used to calculate the PS for each patient. Postoperative complications were analysed with a logistic regression model adjusted according to the PS. Additionally, PS was used to adjust a Cox Model to study disease-free survival (DFS) and overall survival (OS). Kaplan–Meier curves were used to represent oncological outcomes in terms of survival. All analyses were conducted using an intention-to-treat approach, wherein patients in the stent group who ultimately required emergency surgery were analysed within the original stent cohort. Stata^®^ 13.1 (StataCorp, College Station, TX, USA) was used for statistical analysis.

## 3. Results

### 3.1. Patient Characteristics and Operative Data

A total of 50 distinct hospitals participated in this study with a total sample of 2546 patients with pT4 colon cancer (Figure 1). After inclusion and exclusion criteria were applied, a final population of 196 patients was evaluated.

Table 1 summarises patient baseline characteristics and a comparative analysis between the two groups. As shown in Table 1, emergency surgery was performed on 128 patients (65.3%), while 68 (34.7%) received an SEMS as a bridge to surgery. No differences were found in gender distribution, ASA risk, or BMI (body mass index) between the two groups. The median duration of stenting was 12.5 days, interquartile range (6–25.8).

Sigmoidectomy was the most frequent intervention in the SEMS group, in contrast to Hartmann’s procedure in the ES group. In the SEMS group, 54 patients (79.4%) were able to avoid ES and were submitted to elective intervention. Notably, 14 patients (20.6%) from the SEMS group ultimately needed emergency surgery. The reasons for ES were as follows: four tumour perforations, six unresolved obstructions, and four cases lacking data.

Regarding the surgical approach, open surgery was more likely performed in the ES group (122 patients, 95.3%). The creation of an anastomosis was almost eight times more frequent for patients in the SEMS group (OR 7.8).

### 3.2. Postoperative Outcomes

When comparing both groups, variables that demonstrated confounding factors for SEMS placement (age, tumour-related symptoms) were adjusted through a propensity score (PS). Table 2 shows postoperative outcomes and a comparative analysis between the two groups before and after PS adjustment.

ES group patients had more infectious complications (23.1% vs. 13.2%, *p* = 0.1), reaching statistical significance after PS adjustment (*p* = 0.02). Major complications (including reintervention rate) defined as Clavien–Dindo score ≥ 3 were almost 9 times more frequent in the ES group than in the SEMS group (OR 9.6), after PS adjustment. Statistically significant differences were observed when analysing organ-space infection (ES 15.9% vs. SEMS 3.3%, *p* = 0.03). A clinically meaningful outcome metric, optimal result, was obtained in 41 (63.1%) patients in the stent group and 37 (31.6%) patients in the ES group (*p* < 0.01), before and after PS adjustment.

### 3.3. Pathological Tumour Details and Oncological Outcomes

Table 3 presents pathological results and oncological outcomes before and after PS adjustment. No differences were found when comparing adverse histological prognostic factors between the two groups.

The median follow-up was 48.5 months (IC95% 45.3–51.7). A total of 56 patients (28.6%) experienced recurrence at any site in the whole sample, with no statistically significant differences between both groups (*p* = 0.8). Of those, 42 patients (21.4%) presented a systemic recurrence, 23 (11.7%) patients suffered a local recurrence, and 26 patients (13.3%) developed peritoneal metastases during follow-up.

Disease-free survival (DFS) and overall survival (OS) for both groups are shown in Kaplan–Meier curves (Figure 2 and Figure 3). OS and DFS data during 1-year, 3-year, and 5-year intervals are summarised in Table 4. Statistically non-significant differences were found when comparing DFS and OS between groups.

OS and DFS were also analysed considering the hospital size (<200, 200 to 499, and >499 beds). When the Cox model with the PS was adjusted by hospital size, the HR was not modified (HR for DFS varied from 1.40 to 1.36 and OS from 1.60 to 1.60).

When patients who underwent SEMS and emergency surgery due to a clinical or technical failure were compared with patients who underwent ES, the OS (HR = 0.97, 0.34–2.8, *p* = 0.957) and DFS (HR = 0.98, 0.42–2.3, *p* = 0.957) adjusted by PS were similar in both groups.

## 4. Discussion

This study compares postoperative and oncological outcomes of curative colectomy between SEMS and ES groups in patients with pT4 left obstructive colon cancer. The findings demonstrate that SEMSs represent a safe therapeutic strategy, with improved postoperative morbidity (fewer stomas, reduced anastomotic leaks, and lower overall complications) and comparable oncological outcomes. These results align with recent meta-analyses, which show no significant differences in oncological results while highlighting improved postoperative morbidity for SEMS-treated patients [22,23,24,25]. Patients in the SEMS group experienced fewer major complications (Clavien–Dindo ≥ 3), infectious complications, and organ-space complications compared to the ES group. Regarding oncological results, the most controversial point about SEMS insertion, no significant differences were observed in local or distant recurrence, DFS, or OS.

Despite three decades of usage, colonic stent application remains non-standardised across hospital centres and has not been established as the definitive initial management approach for obstructive colon cancer. Early study terminations due to unacceptable perforation rates in the SEMS group generated substantial controversies [8,9,13]. Moreover, these prematurely concluded studies revealed potentially worse recurrence and survival rates in patients experiencing stent insertion perforations.

Contemporary advancements have significantly improved perforation rates and technical success for endoscopic procedures compared to previous publications [26]. Enhanced endoscopic techniques likely reduced waiting times for the beginning of an oncological treatment, potentially explaining improved prognosis compared to previous studies. These improvements justify SEMSs’ inclusion in initial management algorithms for obstructive colon cancer, as recommended by WSES 2017 guidelines and the 2020 ESGE guideline (strong recommendation, high-quality evidence) [15,27].

Stenting time has also been reviewed in guidelines. The median duration of stenting was 12.5 days in our study, which aligns with ESGE recommendations, despite low-quality evidence. The interval between stent insertion and elective surgery represents the balance between stent-related adverse events (reduced by a short interval) and surgical outcomes (improved by a long interval) [15]. Even if there are no prospective studies on this topic, current evidence suggests that a patient’s recovery before elective surgery determines better postoperative results.

pT4 colon tumours have acquired a growing relevance due to their substantially worse oncological prognosis, primarily attributed to increased peritoneal metastasis risk. Previous research identified urgent surgery and tumour perforation as significant prognostic risk factors, when referring to pT4 colon tumours [18]. In our series and in some recent publications that compare SEMSs with ES, there are no differences in perforation rates between the two groups, and even a higher perforation rate was found in the ES group [28,29]. SEMSs appear particularly advantageous for pT4 tumours, potentially mitigating poor prognostic factors by reducing urgent surgeries and stoma creation without increasing tumour perforation risks [30]. Another possible advantage of SEMSs would be the possibility of administering neo-adjuvant treatment. However, SEMS complications could be seen during neo-adjuvant treatment; therefore, this strategy must be further studied before it can be widely recommended.

Unlike other series in which every T stage is included, we subanalysed pT4 tumours, defined as the invasion through the colonic wall and/or into nearby tissues or organs. It is also worth saying that, as pT4 tumours involve all the layers of the colonic wall, they have higher obstruction risks than other tumours with a lower depth of invasion and potentially more challenging stent placement due to increased tumour mass. Despite this, and as we show in our sample, SEMS placement is a safe alternative that allows preoperative recovery and subsequently improves postoperative morbidity, without worsening oncological outcomes.

Regarding the improvement in postoperative complications, SEMS placement allows patient preoperative recovery, in terms of hydroelectrolyte balance readjustment, colonic decompression, and nutritional parameter improvement, thereby reducing urgent surgery-associated morbimortality [7]. Furthermore, as shown in our results, SEMSs allowed a minimally invasive surgery rate of almost 35%, compared to 5% in the ES group.

Reduced postoperative complications may enhance patients’ eligibility for adjuvant treatment, positively influencing oncological outcomes [31]. We also introduce the variable ‘optimal result’, defined as a successful surgery without major postoperative morbimortality or stoma. An optimal result would summarise the best way to manage patients with obstructive colon cancer, which should be pursued by every surgeon. An optimal result was more often obtained when patients were treated with SEMSs in our sample.

Our study joins others recently published with similar results [31,32,33,34,35], demonstrating that colonic stenting is a good alternative to urgent surgery for the management of obstructive colon cancer, improving postoperative results without worsening oncological results. We provide novel insights with our series, showing that obstructive large tumours related to the worst prognosis (pT4) can be safely managed with SEMS placement.

There is plenty of literature that endorses SEMS placement as the initial approach for obstructive colon cancer, although there are still some questions that remain unresolved. The optimal interval between SEMS placement and elective surgery is yet to be defined.

The success of the SEMS placement technique raises important considerations about centralisation. Instead of implementing the procedure universally, we should maybe consider establishing referral channels to tertiary hospitals with advanced endoscopic capabilities and specialised expertise. Considering that approximately 30% of colon cancer cases present with obstruction, directing patients to centres with sophisticated equipment and highly qualified endoscopists could significantly improve patient management and technical outcomes.

Centralisation strategies, previously demonstrated effective in managing other malignancies like gastric cancer [36], may represent an appropriate approach for patients with obstructive colon cancer. We understand that our study has certain limitations inherent to a retrospective and multicentre study. We experienced data loss despite rigorous data collection. Besides that, data collection was initially focused on pT4 tumour management, so there is a lack of information regarding some specific SEMS placement issues (technical or clinical failure). We also must mention the variability in treatment among different centres. Our data were obtained from a larger sample of patients, so we have a lack of information on whether all the participant centres had the possibility for SEMS placement. Stent placement is decided individually in each case. Information regarding whether SEMS was considered technically or clinically feasible is missing, which represents a selection bias. Data regarding clinical conditions that led to the decision-making such as patients’ clinical status, colorectal specialised surgical team, tumour size, or stent availability are also missing, representing an important limitation of our study.

Despite these limitations, this study’s strengths are substantial. By enrolling 50 different hospitals, we achieved an acceptable sample size, capturing diverse clinical approaches and giving value to the data. Also, this manuscript presents an analysis of a specific group of patients, pT4 colon cancer patients presenting with acute bowel obstruction. Finally, a propensity score analysis was performed to decrease the impact of confounding factors that are inherent to retrospective studies.

Some practical implications can be drawn from our study: SEMS placement is a safe strategy to be considered as the initial approach for obstructive colon cancer, even for pT4 tumours. Moreover, stent placement should be performed by experienced clinicians, to obtain successful results. Given that, maybe we should consider enabling referral channels to tertiary hospitals where this technique can be performed if it is not available in our centre.

## 5. Conclusions

Self-expanding metallic stents represent a good alternative for the management of patients with pT4 obstructive left colon cancer, when technically feasible. This approach is a good alternative to emergency surgery, improving postoperative outcomes without worsening short- and medium-term oncological results.

## Figures and Tables

**Figure 1 cancers-16-04096-f001:**
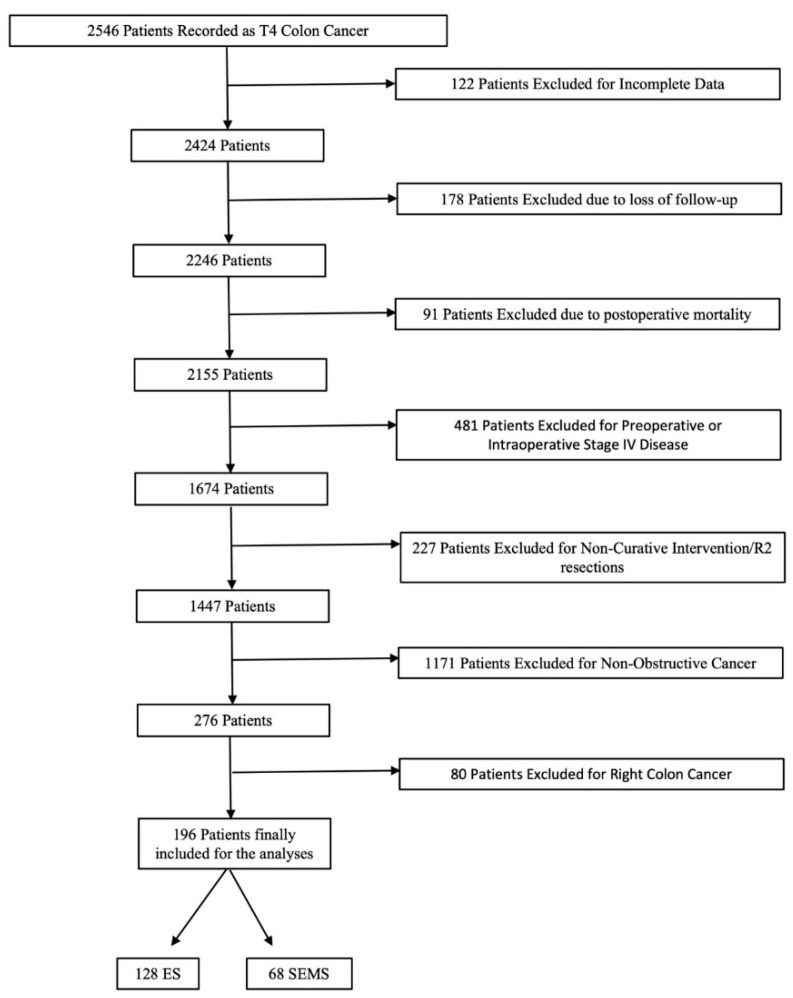
Flowchart detailing the selection of the patients in this study.

**Figure 2 cancers-16-04096-f002:**
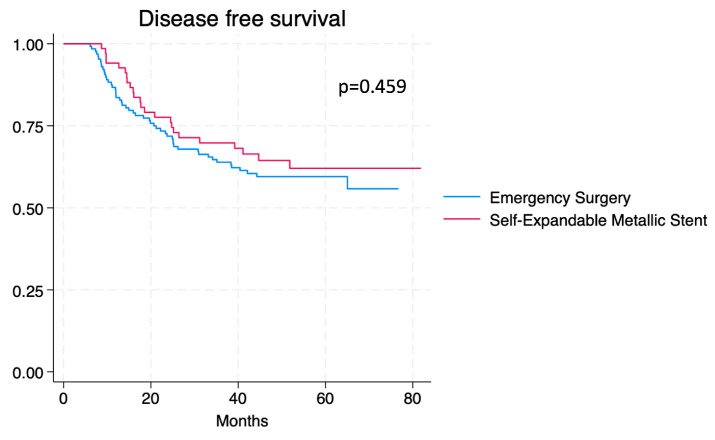
Kaplan–Meier curve for PDFS comparing patients who received emergency surgery vs. SEMS.

**Figure 3 cancers-16-04096-f003:**
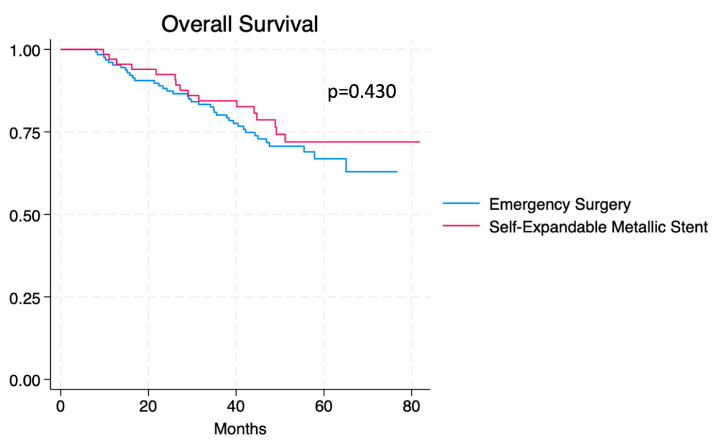
Kaplan–Meier curve for OS comparing patients who received emergency surgery vs. SEMS.

**Table 1 cancers-16-04096-t001:** Baseline characteristics and comparison between two groups.

	ES *n* = 128	SEMS *n* = 68	Total *n* = 196	*p*	OR (IC95%)
**Age (years), mean (SD)**	66.1(12.5)	73.4 (11.2)	69 (12.4)	<0.01	1.04 (1.02–1.08)
**Gender** Male Female	76 (59.4%) 52 (40.6%)	39 (57.4%) 29 (42.7%)	115 (58.7%) 81 (41.3%)	0.78	1.08 (0.6–2)
**ASA** I-II III-IV	73 (58%) 53 (42%)	36 (53%) 32 (47%)	109 (56%) 85 (44%)	0.5	0.8 (0.5–1.5)
**BMI (Kg/m^2^)** <30 >30	66 (72.5%) 25 (27.5%)	39 (72.2%) 15 (27.8%)	105 (72.4%) 40 (27.6%)	0.97	1.01 (0.48–2.2)
**Surgical scheduling** Elective surgery Emergency surgery	0 (0%) 128 (100%)	54 (79%) 14 (20.6%)	54 (27.6%) 142 (72.4%)	<0.01	-
**Free tumour** Yes No	90 (72.6%) 34 (27.4%)	44 (65.7%) 23 (34.3%)	134 (70.2%) 57 (29.8%)	0.32	0.72 (0.38–1.37)
**Type of surgery** Left colectomy Sigmoidectomy Hartmann Others	26 (20.6%) 31 (24.6%) 38 (30.2%) 31 (24.6%)	15 (22.4%) 41 (61.2%) 4 (5.9%) 7 (10.5%)	41 (21.2%) 72 (37.3%) 42 (21.8%) 38 (19.7%)	<0.01	2.6 (0.9–7.2) 5.9 (2.3–15) 0.5 (0.1–1.7)
**Stoma** Yes No	70 (58.3%) 50 (41.7%)	13 (20.3%) 51 (79.7%)	83 (45%) 101 (54.9%)	<0.01	0.2 (0.1–0.4)
**Surgical approach** Open surgery MIS	122 (95.3%) 6 (4.7%)	45 (66.2%) 23 (33.8%)	167 (85.2%) 29 (14.8%)	<0.01	10.4 (3.9–27.2)
**Adjuvant chemotherapy**	98 (76.6%)	45 (66.2%)	143 (73%)	0.1	1.7 (0.9–3.2)

ES: emergency surgery, SEMS: self-expandable metallic stent, ASA: American Society of Anesthesiologists, BMI: body mass index; MIS: minimally invasive surgery. OR: odds ratio; IC-95%: 95% confidence interval; free tumour: no adhesions to adjacent viscera, abdominal wall, or peritoneum.

**Table 2 cancers-16-04096-t002:** Postoperative outcomes before and after PS adjustment.

	Before PS Adjustment				After PS Adjustment	
	ES *n* = 128	SEMS *n* = 68	*p*	OR/HR (CI95%)	*p*	OR (CI95%)
**Operation-Related Outcomes**						
Any complication	76 (60.8%)	35 (53%)	0.3	1.4 (0.8–2.5)	0.08	1.8 (0.9–3.4)
Infectious complication	27 (23.1%)	9 (13.2%)	0.1	1.9 (0.9–4.5)	0.02	3 (1.2–7.3)
Organ-space infection	17 (15.9%)	2 (3.3%)	0.03	5.6 (1.2–25)	<0.01	9.3 (2–43.9)
Anastomotic leak	6 (13%)	3 (5.8%)	0.2	2.6 (0.6–11.3)	0.06	4.7 (0.9–24.4)
Perioperative transfusion	33 (28.5%)	9 (16.7%)	0.1	2 (0.9–4.5)	0.05	2.3 (1–5.5)
Major complication (CD ≥ 3)	28 (22.4%)	3 (4.5%)	<0.01	6.2 (1.8–21.1)	<0.01	9.6 (2.7–34.6)
Stoma creation	70 (58.3%)	13 (20.3%)	<0.01	5.5 (2.7–11.2)	<0.01	7 (3.2–15.1)
Optimal result	37 (31.6%)	41 (63.1%)	<0.01	3.7 (2–7)	<0.01	0.2 (0.1–0.4)

ES: emergency surgery, SEMS: self-expandable metallic stent, PS: propensity score, OR: odds ratio, HR: hazard ratio, CD: Clavien–Dindo.

**Table 3 cancers-16-04096-t003:** Pathological results and oncological outcomes.

	Before PS Adjustment				After PS Adjustment	
	ES *n* = 128	SEMS *n* = 68	*p*	OR/HR (IC95%)	*p*	OR/HR (IC95%)
**Pathological Results**						
Perineural invasion	48 (37.5%)	25 (37.9%)	0.9	0.9 (0.5–1.8)	0.8	0.9 (0.5–1.7)
Lymphatic invasion	50 (39%)	33 (50%)	0.1	0.6 (0.4–1.2)	0.2	0.6 (0.3–1.2)
Vascular invasion	54 (42.2%)	31 (47%)	0.5	0.8 (0.5–1.5)	0.5	0.8 (0.4–1.5)
Tumour perforation	27 (21%)	10 (14.7%)	0.3	1.6 (0.7–3.4)		
Number of lymph nodes - Resected number - N0 - N+	24.3 (SD 19.4) 57 (44.5%) 71 (55.5%)	23.2 (SD 12.5) 32 (47%) 36 (52.9%)	0.7 0.7 0.7	1.1 (−4–6.2) 1.1 (0.6–2)	0.8 0.9	−0.7 (−6–4.6) 0.9 (0.5–1.8)
**Oncological Outcomes**						
Recurrence	36 (28.1%)	20 (29.4%)	0.8	0.9 (0.5–1.8)	0.9	0.9 (0.5–1.9)
Disease-Free Survival (months)	44.3	44.5	0.5	1.2 (0.7–1.9)	0.2	1.4 (0.8–2.3)
Overall Survival (months)	47.6	50.5	0.4	1.3 (0.7–2.3)	0.1	1.6 (0.9–2.9)

ES: emergency surgery, SEMS: self-expandable metallic stent, PS: propensity score, OR: odds ratio, HR: hazard ratio, SD: standard deviation.

**Table 4 cancers-16-04096-t004:** Overall survival and disease-free survival at 1–3–5 years.

	ES *n* = 128	SEMS *n* = 68
Overall Survival (OS)		
1-year	122 (95.3%)	66 (97%)
3-year	100 (78.1%)	53 (77.9%)
5-year	24 (18.8%)	14 (20.6%)
Disease-Free Survival (DFS)		
1-year	110 (85.9%)	66 (97%)
3-year	81 (63.3%)	45 (66.2%)
5-year	22 (17.2%)	10 (14.7%)

ES: emergency surgery; SEMS: self-expandable metallic stent.

## Data Availability

The data will be accessible to researchers who make reasonable requests to the authors.

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
