# Peer review of "Beyond Obstruction: Evaluating Self-Expandable Metallic Stents (SEMSs) vs. Emergency Surgery for Challenging pT4 Obstructive Colon Cancer: Multicentre Retrospective Study"

_cancers, 2024, doi:10.3390/cancers16234096_

Round 1
Reviewer 1 Report
Comments and Suggestions for Authors
very interesting multicenter retrospective study comparing emergency surgery in left colon adenocarcinoma occlusion and positioning of a self-expanding prosthesis to postpone the resective intervention to a second time. We agree with the mortality rates reported by colleagues, especially because often the occluded patients are elderly, affected by comorbidities and arrive in the emergency room already in critical conditions with important metabolic and hydroelectric imbalances. We agree with the decision to proceed with surgery or prosthetic positioning after an imaging study that allows us to highlight the T of the neoplasm but also M and N. Another data to take into close consideration is the distension upstream of the obstruction that conditions the prosthetic positioning. Often we have to fear more the displacement of the prosthesis rather than the perforation of the viscus at least as far as our experience is concerned. We absolutely agree with the use of the prosthesis only in the left and sigma colon. The primary and secondary endpoints are interesting and shareable. The cohort of patients enrolled is made up of a fair number of people for statistical processing, on which we have nothing to object and to be able to elaborate a discussion. We agree with colleagues on the undoubted advantages of prosthetic positioning in the terms described and we agree that there are still some hesitations on this. But to obtain the best results we must have high-volume centers with the necessary experience, (doi.org/10.3390/jcm12072708 to be cited in the bibliography). It is understandable that there are no differences in the prognosis of the disease, but if we look at the process from diagnosis to the beginning of an oncological treatment, the times are certainly shortened and this can also have repercussions on the prognosis. We know that the reduction of waiting times for the beginning of an oncological treatment is fundamental in all oncology. Furthermore, rightly, prosthetic positioning can make minimally invasive surgery possible with undoubted advantages. We agree on the weak points of the paper. Excellent iconography, good English, Good bibliography
Author Response
We thank you for the analysis and the comments about the manuscript entitled: Beyond Obstruction: Evaluating Self-Expandable Metallic Stents (SEMS) vs. Emergency Surgery for Challenging pT4 Obstructive Colon Cancer: Multicentre Retrospective Study.
We feel confident that you gave our manuscript the very best peer review and we are pleased to respond to the comments of your referees.
We have responded to the Reviewers’ comments and revised all the manuscript introducing changes as suggested by the editor and the reviewers.
The remarks incorporated in the text are highlighted in yellow.
- QUESTION: Very interesting multicenter retrospective study comparing emergency surgery in left colon adenocarcinoma occlusion and positioning of a self-expanding prosthesis to postpone the resective intervention to a second time. We agree with the mortality rates reported by colleagues, especially because often the occluded patients are elderly, affected by comorbidities and arrive in the emergency room already in critical conditions with important metabolic and hydroelectric imbalances. We agree with the decision to proceed with surgery or prosthetic positioning after an imaging study that allows us to highlight the T of the neoplasm but also M and N. Another data to take into close consideration is the distension upstream of the obstruction that conditions the prosthetic positioning. Often we have to fear more the displacement of the prosthesis rather than the perforation of the viscus at least as far as our experience is concerned. We absolutely agree with the use of the prosthesis only in the left and sigma colon. The primary and secondary endpoints are interesting and shareable. The cohort of patients enrolled is made up of a fair number of people for statistical processing, on which we have nothing to object and to be able to elaborate a discussion.
We agree with colleagues on the undoubted advantages of prosthetic positioning in the terms described and we agree that there are still some hesitations on this. We agree on the weak points of the paper. Excellent iconography, good English, Good bibliography
RESPONSE: thank you very much for your support and constructive review.
- QUESTION: But to obtain the best results we must have high-volume centers with the necessary experience, (doi.org/10.3390/jcm12072708 to be cited in the bibliography).
RESPONSE: we have added this reference about the importance of high-volume centers in the discussion (second paragraph in page 13):
Centralization in high volume centers has already been proved to be a good option for other pathologies, such as gastric cancer [36]; therefore, this strategy could be appropriate for patients with obstructive colon cancer.
- QUESTION: It is understandable that there are no differences in the prognosis of the disease, but if we look at the process from diagnosis to the beginning of an oncological treatment, the times are certainly shortened and this can also have repercussions on the prognosis. We know that the reduction of waiting times for the beginning of an oncological treatment is fundamental in all oncology. Furthermore, rightly, prosthetic positioning can make minimally invasive surgery possible with undoubted advantages.
RESPONSE: we agree that the shortened time for oncological treatment could be an important point to improve the prognosis. We have included this point in the discussion (page 11, last paragraph):
The improvement in the endoscopic techniques could have shortened the waiting time for the beginning of an oncological treatment and this could explain the improved prognosis compared to previous studies. All these improvements explain that SEMS has been…
Reviewer 2 Report
Comments and Suggestions for Authors
Thank you for allowing me to review this retrospective multicenter study comparing the results of first surgery and those of insertion of a colic porthesis followed by surgery in the context of T4 left colon cancers in occlusion. the inclusion and exclusion criteria are clearly stated and the methodology by propensity score robust.
Did the authors take into account the influence of the center and the operative volume in their propensity score?
severe morbidity is significantly higher in the first surgery group. in some countries, it is recommended to perform an immediate preoperative stoma in emergency followed by elective surgery as in patients initially treated with stent. Did the authors look at the results of this two-step strategy in their cohort?
what was the oncological outcome of patients (21) with stent failure who had to be operated on urgently.
in the discussion I suggest to the authors to discuss the place of the neo-adjuvant treatment within these T4 in occlusion that are derived.
Author Response
We thank you for the analysis and the comments about the manuscript entitled: Beyond Obstruction: Evaluating Self-Expandable Metallic Stents (SEMS) vs. Emergency Surgery for Challenging pT4 Obstructive Colon Cancer: Multicentre Retrospective Study.
We feel confident that you gave our manuscript the very best peer review and we are pleased to respond to the comments of your referees.
We have responded to the Reviewers’ comments and revised all the manuscript introducing changes as suggested by the editor and the reviewers.
The remarks incorporated in the text are highlighted in yellow.
We have addressed the Editorial Requirements and we resubmit the revised paper for your reconsideration for publication in Cancers.
- QUESTION: Thank you for allowing me to review this retrospective multicenter study comparing the results of first surgery and those of insertion of a colic porthesis followed by surgery in the context of T4 left colon cancers in occlusion. the inclusion and exclusion criteria are clearly stated and the methodology by propensity score robust.
RESPONSE: thank you very much for your support and constructive review.
- QUESTION: Did the authors take into account the influence of the center and the operative volume in their propensity score?
ANSWER: we have performed this analysis. When DFS and OS was adjusted for both PS and hospital size, the HR was not modified. This information has been added to the results (last paragraph, page 9):
OS and DFS was also analyzed considering the hospital size (<200, 200 to 499 and >499 beds). When the Cox model with the PS was adjusted by hospital size the HR was not modified (HR for DFS varied from 1.40 to 1.36 and OS from 1.60 to 1.60).
- QUESTION: Severe morbidity is significantly higher in the first surgery group. in some countries, it is recommended to perform an immediate preoperative stoma in emergency followed by elective surgery as in patients initially treated with stent. Did the authors look at the results of this two-step strategy in their cohort?
ANSWER: this is not a common practice in our country. In our environment stomas are only used for rectal cancer. Colon cancers with obstruction are submitted to either resection surgery or stent placement; therefore, assessing the use of stoma in this setting is not possible in our. cohort.
- QUESTION: what was the oncological outcome of patients (21) with stent failure who had to be operated on urgently.
ANSWER: in the las paragraph of the results (page 10) we have included this analysis:
When patients who underwent SEMS and urgent surgery due to a clinical o technical failure were compared with patients who underwent ES, the OS (HR=0.97, 0.34-2.8, p=0.957) and DFS (HR=0.98, 0.42-2.3, p=0.957) adjusted by PS was similar in both groups.
- QUESTION: In the discussion I suggest to the authors to discuss the place of the neo-adjuvant treatment within these T4 in occlusion that are derived.
ANSWER: we fully agree with the reviewer that SEMS could also allow nea-adjyvant treatment. We have included this discussion (3rd paragraph of page 12):
Another possible advantage of SEMS would be the possibility of administering neo-adjuvant treatment. However, SEMS complications could be seen during neo-adjuvant treatment; therefore, this strategy must be further studied before it can be widely recommended.
Round 2
Reviewer 2 Report
Comments and Suggestions for Authors
the authors responded point by point to comments and questions designed to improve the quality of the manuscript